# Breakfast consumption patterns and associated factors among adolescent high-school students in Tullo District, Eastern Ethiopia

**Natnael Teferi, Tara Wilfong, Dawit Firdisa ⓘ\*, Samrawit Berihun, Behailu Hawulte**

School of Public Health, College of Health and Medical Sciences, Haramaya University, Harar, Ethiopia

\* firdisadawit@gmail.com

## Abstract

### Background

There is growing proof to recommend eating breakfast has positive health and school-related outcomes for adolescents, including improved performance, attention, brain development, and physical growth. However, there is a dearth of evidence on the comprehensive understanding of breakfast consumption patterns and associated factors. Therefore, this study aimed to assess breakfast consumption patterns and their associated factors among adolescent high school students in the Tullo district, Eastern Ethiopia.

### Methods

An institution-based cross-sectional study design was conducted among 405 randomly selected adolescent high school students in the Tullo District, Eastern Ethiopia, from October 09–29, 2023. A self-administered questionnaire was utilized to collect the data. Epidata version 4.6 and SPSS Statistics version 27.0.1 were used for data entry and analysis, respectively. Both bivariable and multivariable logistic regression analyses were performed to identify the factors associated with breakfast consumption patterns. An adjusted odds ratio (AOR) with a 95% confidence interval (CI) was calculated to determine the strength of the association, and a p-value of 0.05 was used to determine statistical significance.

### Result

Nearly half, 46.2% (95% CI: 41.5, 51.4), of participants had irregular breakfast consumption (skipped). Being female (AOR = 5.28; 95% CI: 2.69, 10.36), family size of >5 (AOR = 4.76; 95% CI: 2.41, 9.36), being a rural resident (AOR = 3.34; 95% CI: 1.78, 6.25), no formal maternal education (AOR = 3.89; 95% CI: 2.09, 7.22), chewing khat (AOR = 3.13; 95% CI: 1.59, 6.16), cigarette smoking (AOR = 3.06; 95% CI: 1.02,

**Data availability statement:** All relevant data are within the manuscript and its Supporting Information files.

**Funding:** The author(s) received no specific funding for this work.

**Competing interests:** The authors declare that they have no conflicts of interest.

**Abbreviations:** AOR, Adjusted Odds Ratio; BAZ, BMI-for-age-Z-score; BMI, Body Mass Index; CDC, Center for Disease Control; CED, Chronic Energy Deficiency; CSA, Central Statistical Agency; EDHS, Ethiopian Demographic Health Survey; FANTA, Food and Nutrition Technical Assistance; GBD, Global Disease Burden; HBSC, Health Behavior in School-aged Children; IHRERC, Institutional Health Research Ethics Review Committee; MUAC, Mid Upper Arm Circumference; NGO, Non-Governmental Organization; SCOFF, Sick, Control, One, Fat, Food; SS, Systematic sampling; T2DM, Type 2 diabetes mellitus; TEM,Technical Error of Measurement; UNICEF, United Nations International Children's Emergency Fund; WFP, World Food Program; WHO, World Health Organization.

9.17), and eating disorders (AOR = 6.54; 95% CI: 2.19, 19.43) were significantly associated with irregular breakfast consumption patterns among adolescents.

## Conclusion

The findings of this study showed that the prevalence of irregular breakfast consumption (breakfast skipping) among adolescent high school students was high. Being female, rural residency, no formal maternal education, current smoking of cigarettes, current khat chewing, and eating disorders were identified as factors associated with breakfast consumption patterns. Given that almost half of adolescents in Tullo District skip breakfast, several modifiable factors associated with this practice, focused interventions are essential.

## Introduction

Breakfast is the primary and most important meal of the day which is considerably related to physiological, psychological, and social well-being. [1,2]. A healthy breakfast contains a balanced ratio of all the required nutrients for our body, and it should guarantee a median of 20–25% of the energy consumed throughout the day. [3]. Breakfast consumption improves attention, memory, and psychological performance. [4]. Eating breakfast is notably necessary during the adolescent period. Adolescents go through different biological, physical, and mental development processes. [5]. Moreover, during this period, adolescents have the greatest total energy requirement compared to any age group (~2,420 kcal/day) [6]. Therefore, breakfast consumption contributes approximately 24.3% to 27.8% of daily caloric intake, averaging around 680.7 kcal. [7].

Globally, the prevalence of breakfast skipping among adolescents ranged between 0.7% (in Japan) and 94% (in Portugal) [8–10]. In low- and middle-income countries (LMICs), the prevalence of breakfast skipping ranges between 23% and 38% [11,12]. Moreover, studies conducted in different parts of Ethiopia reported that breakfast skipping has ranged between 20% and 42% [4,13,14].

Missing breakfast is associated with risk factors for cardiac and metabolic health problems and is important in weight management. [15]. Similarly, several studies have shown that missing breakfast is related to worse lipid profiles, vital sign levels, endocrine resistance, and metabolic syndrome. [16]; increased risk of type 2 DM (diabetic mellitus) [17]; higher BMI (body mass index) [18]; and reduced performance in cognitive and psychosocial functions, as well as academic learning and achievement [13].

Breakfast consumption patterns are associated with sociodemographic, behavioral, and environmental factors. Sociodemographic factors include socioeconomic status, age, sex, and type of school. [19,20]. Environmental factors include eating or buying food prepared outside the home, maternal education and employment, and parental death. [11,19]. Breakfast skipping is significantly associated with an increased risk of type 2 DM. [17] poor academic performance, depression, lower

happiness, posttraumatic stress disorder, loneliness, short and long sleep, sleep problems, restless sleep, and increasing levels of family income [21,22]. In Ethiopia, studies have shown that breakfast skipping among adolescents is associated with poorer academic performance. [4,13], and obesity [23]. Adolescents from low-income families and food-insecure households tend to skip their breakfast more than their counterparts; being female is also an important factor in skipping breakfast. [13]. This age group was chosen because, as they transition to adulthood, they are more likely to be left to set food plans, including breakfast consumption, than younger children [24].

Different scholars have intensively studied factors associated with breakfast consumption patterns, and the nutritional status of adolescents in Ethiopia [4,13,14]. However, there is a lack of clear data or insufficient studies on adolescent nutrition, in particular breakfast consumption patterns. Moreover, no study has been conducted particularly in this study area. Therefore, this study aimed to assess breakfast consumption patterns and associated factors among high school adolescent students in the Tullo district, Eastern Ethiopia.

## Methods and materials

### Study design and period

An institution-based cross-sectional study design was conducted. The study was conducted from October 09–29, 2023, in the Tullo district, Eastern Ethiopia. Tullo Woreda is one of the 17 Woredas of the West Hararghe Zone of the Oromia Region and is located 42 km from the zonal capital, Chiro, and 244 km from the capital of Addis Ababa. As per the 2007 population projection, the district has a total population of 147,384. The woreda covers 485.84 sq. km. The majority of the household's income depends on small-scale cash cropping, mainly coffee (70.7%). According to the District Education office report, currently, there are four high schools with 3983 total high school students (Tullo woreda; Health office, Education office, and Agricultural office, 2021).

### Population and sampling

The source population for this study was all adolescents attending high schools in the Tullo District of Eastern Ethiopia. All randomly selected adolescents from the selected high schools during the study period in the Tullo District, Eastern Ethiopia, were the study population. All adolescents aged 14–19 years, registered during the academic year of 2023–2024 from selected high schools in the Tullo district, were included in the study, whereas students who had visual impairment and were critically ill during data collection were excluded from our study.

The required sample size was calculated by using a single population proportion formula. The proportion (p) of breakfast skipping is estimated to be 42.3% among adolescents from a previous study conducted on the assessment of breakfast eating habits and their association with cognitive performance of early adolescents [25]. By considering a 10% non-response rate, the sample size was estimated to be 414. We chose two out of the four public high schools in Tullo District using a simple random sampling method (lottery). A proportional stratified sampling technique was employed. The total sample size was allocated to the schools and sections proportional to the number of students in each selected school and section at the time of the study. The study participants were selected by a systematic sampling technique using the list of students enrolled in each school and section as a sampling frame. The sampling interval was determined by dividing the total number of students in the respective school grade level by the allocated sample size and was found to be five. The first participant was selected randomly by the lottery method, and then every fifth adolescent student was included in the study.

### Data collection and quality control

A pretested structured questionnaire was used to collect data in four sections: Section A on sociodemographic and socioeconomic status (SES), which were assessed using questions adapted from the Ethiopia Demographic and Health

Survey 2016 report [26]; Section B on lifestyle factors like health status, smoking, alcohol use, and sleep. For assessment of eating disorders, the SCOFF(Sick, Control, One, Fat, Food) questionnaire validated among adolescents was used [27]. An adolescent having a score equal to or above 2 was considered to be at risk for eating disorders. Section C on breakfast patterns, including items consumed and reasons which was assessed using a standardized food frequency questionnaire [28]. There were a total of 16 food items or categories on the questionnaire; these items were then categorized into 9 main food groups by summing all the consumption frequencies of food items of the same group and recoding the value of each group above 7 as 7. Next, to construct new weighted food group scores, multiply the value acquired for each food group by its weight (Main Staples * 2, Pulses * 3, Vegetables * 1, Fruit * 1, Meat and Fish * 4, Milk * 4, Sugar * 0.5, Oil * 0.5, and Condiments * 0). The food consumption score was then calculated by adding the weighted food category scores and applying the appropriate thresholds (0–21 Poor, 21.5–35 Borderline, and > 35.5 Borderline) [29]. Section D on anthropometric measurements assessed participants' nutritional status. The questionnaire was prepared in English and then translated into Afan Oromo and retranslated to English in order to check for consistency.

Data was collected by three BSC holder nurses fluent in the local language. The interview was conducted using a pretested questionnaire over 20 days. The process included self-reported responses and physical measurements. Weight and height scales were calibrated to ensure reliability. Students removed their shoes for weight, recorded to the nearest 0.1 kg, and stood barefoot for height, measured to the nearest 0.5 cm. Body mass index (BMI) was calculated (kg/m$^2$), and WHO 2007 growth charts classified BMI-for-age-Z-score (BAZ) using z-scores for those aged 5–19 [30].

To maintain the quality and consistency of the data collection, two days of training were given to data collectors. This covered the objective of the study, procedures, and ethical issues. A pretest was conducted on 5% of the sample size outside the study area. Data collected were checked daily for completeness and consistency by the principal investigators. The scales were regularly checked and adjusted to zero after each measurement. To minimize measurement error, TEM (Technical Error of Measurement) was done before actual data collection with ten participants, and acceptable values for intra-evaluator and inter-evaluator were less than 1.5% and 2%, respectively.

### Operational definitions

**Adolescents.** According to WHO definitions, adolescents are the age group between 10 and 19 years old [31].

**Breakfast.** Defined as the first meal of the day, eaten before starting daily activities, before 10:00 am [13].

**Breakfast consumption patterns.** Those who ate breakfast every day or six days in the 7 days preceding the survey were categorized as regular breakfast consumers, and those who skipped breakfast more than 2 days were categorized as breakfast skippers [4].

**Food consumption score.** Used for household food security, calculated by examining the number of times certain foods, grouped into basic food groups, food consumed score was categorized as: (0–21) Poor; (21.5–35) Borderline; (>35) Acceptable. [29].

**BMI.** The World Health Organization 2007 growth reference was used as a standard reference for classifying adolescents based on body mass index for age. BMI for age was classified as <−2 underweight, between −2 and 2 is normal, and >1 is overweight [30].

**Family size.** Refers to the total number of people living in a house during the study period, and respondents are categorized as small family size (≤5) and large family size (>5) [13].

**Eating disorder.** The questionnaire consists of five questions, and the response options ('Yes'/'No') were scored by giving one point for a positive answer and zero points for a negative answer, and respondents who scored equal to or above 2 were considered to be at risk for eating disorders [27].

## Data analysis

After checking for completeness and consistency, the data were exported to SPSS Statistics version 27.0.1 for cleaning and analysis. Descriptive analysis was performed, and results were presented in tables, graphs, and charts. A bivariate logistic regression analysis was performed to assess associations between the dependent variable and the independent variables. The independent variables with p-values less than 0.25 in the bivariate analysis were considered for further analysis by multivariate logistic regression analysis to control for potential confounders and to detect the predictors of breakfast consumption patterns. Odds ratios, along with a 95% confidence interval, were estimated to measure the strength of the association. The level of statistical significance was declared at a P value less than 0.05. Multicollinearity was checked by calculating VIF, and model adequacy was checked by using the Hosmer and Lemeshow goodness of fit test.

## Ethics approval and informed consent

The study was conducted according to the Declaration of Helsinki. Ethical clearance was obtained from the Institutional Health Research Ethics Review Committee (IHRERC) of the College of Health and Medical Sciences, Haramaya University, with reference number IHRERC/181/2023. An official letter was written to the Tullo Woreda Education office for cooperation. A letter of permission was obtained from the Tullo Woreda Education office and selected high schools before the study was conducted. Informed, voluntary, written, and signed consent was obtained from the adolescent's family for those less than 18 years of age and from students themselves if over 18 years old after informing them about the purpose, risks, and benefits of the study.

## Results

### Socio-demographic characteristics

From 414 eligible participants, 405 completed the study, yielding a response rate of 97.8%. More than half, 53.6%, of the participants were from grade 10, and 54.6% were female. The mean age of the study participants was 16.63, and the SD was 1.31. From all participants' parents, 72.1% of the fathers and 42.2% of the mothers had attained some level of formal education (Table 1).

### Behavioral and medical characteristics

According to this study, 43.2% were currently khat chewers, 14.8% were cigarette smokers, and 5.9% consumed alcoholic drinks in the past 30 days. Regarding the time of wake-up, 37.0% of participants usually awoke before 6:00 am, and 2.7% of participants were awake after 8:00 a.m. Among the participants, 36.3% usually slept after 11:00 p.m., and 14.3% slept before 9:00 p.m. 12.1% of the total participants had an eating disorder or at least one symptom. Of those, 6.4% reported they make themselves sick because they feel uncomfortably full. The majority of respondents, 89.60%, walk to school, and only 10.4% use transport (Table 2).

### Nutritional characteristics

Based on nine food groups, a food consumption score was calculated, and 69.6% had food consumption that could be classified as acceptable. Based on the BMI for age, 16.8% were underweight and 80% were within the normal range. More than three-fourths, 77.5% of participants, never ate between meals during the day, and 66.40% of participants ate less than three meals a day (Table 3).

### Breakfast consumption patterns

With regard to breakfast consumption patterns, 46.2% skipped breakfast (95% CI: 41.5, 51.4), and 53.8% of adolescents regularly consumed breakfast. A chi-square test of independence to evaluate the relationship between sex and breakfast

**Table 1. Socio-demographic characteristics of adolescent high school students in the Tullo district, Oromia Region, Eastern Ethiopia, 2023.**

| Variables | Category | Frequency (405) | Percentage (%) |
|---|---|---|---|
| Age (in year) | Middle adolescent (14 –16 ) | 78 | 19.3 |
| | Late adolescent (17 –19) | 327 | 80.7 |
| Sex | Female | 221 | 54.6 |
| | Male | 184 | 45.4 |
| Education level | Grade 9 | 188 | 46.4 |
| | Grade 10 | 217 | 53.6 |
| Religion | Orthodox | 103 | 25.4 |
| | Muslim | 272 | 67.2 |
| | Protestant | 24 | 5.9 |
| | Catholic | 6 | 1.5 |
| Residence | | | |
| | Urban | 227 | 56.0 |
| | Rural | 178 | 44.0 |
| Number of family members | ≤5 | 150 | 37.0 |
| | >5 | 255 | 63.0 |
| With whom respondent live currently | Alone | 39 | 9.6 |
| | With friends | 130 | 32.1 |
| | With family | 236 | 58.3 |
| Mother's marital status | Single | 106 | 26.2 |
| | Married | 227 | 56.0 |
| | Divorced | 57 | 14.1 |
| | Widowed | 15 | 3.7 |
| Mother's educational status | No formal education | 234 | 57.8 |
| | Primary school | 130 | 32.1 |
| | Secondary school | 18 | 4.4 |
| | College and above | 23 | 5.7 |
| Father's educational status | No formal education | 113 | 27.9 |
| | Primary school | 120 | 29.6 |
| | Secondary school | 103 | 25.4 |
| | College and above | 69 | 17 |
| Mother's occupation | House wife | 161 | 39.8 |
| | Merchant | 86 | 21.2 |
| | Farmer | 140 | 34.6 |
| | Government employer | 18 | 4.4 |
| Father's occupation | Government employer | 53 | 13.1 |
| | Merchant | 86 | 21.2 |
| | Farmer | 262 | 64.7 |
| | Others** | 4 | 1.00 |
| Average monthly income | >2000ETB | 298 | 73.6 |
| | < 2000ETB | 107 | 26.4 |
| Living arrangement | Alone | 39 | 9.6 |
| | With friends | 130 | 32.1 |
| | With family | 236 | 58.3 |

Others**Private employee, NGO, pension*

**Table 2. Behavioral and medical characteristics of adolescent high school students in Tullo district, Oromia Region, Eastern Ethiopia 2023.**

| Variables | Category | Frequency (405) | Percentage (%) |
|---|---|---|---|
| Current khat use | Yes | 175 | 43.2 |
| | No | 230 | 56.8 |
| Current cigarette smokers | Yes | 60 | 14.8 |
| | No | 345 | 85.2 |
| Current alcohol use | Yes | 24 | 5.9 |
| | No | 381 | 94.1 |
| Usual time of wake-up | Before 6:00 am | 150 | 37.0 |
| | 6:00–7:00 am | 103 | 25.4 |
| | 7:00–8:00 am | 141 | 34.8 |
| | After 8:00 am | 11 | 2.7 |
| Usual time of sleeping for the night | Before 9:00 pm | 58 | 14.3 |
| | 9:00–10:00 pm | 82 | 20.2 |
| | 10:00–11:00 pm | 118 | 29.1 |
| | After 11:00 pm | 147 | 36.3 |
| How do you come to the school | Walking | 363 | 89.6 |
| | Using transports | 42 | 10.4 |
| Having eating disorder | Yes | 49 | 12.1 |
| | No | 356 | 87.9 |

**Table 3. Nutritional characteristics of adolescent high school students in Tullo district, Oromia Region, Eastern Ethiopia, 2023.**

| Variables | Category | Frequency (405) | Percentage (%) |
|---|---|---|---|
| Meal frequency | < 3times a day | 269 | 66.4 |
| | ≥ 3times a day | 136 | 33.6 |
| Snack frequency | One times a day | 63 | 15.6 |
| | Two times a day | 28 | 6.6 |
| | Never | 314 | 77.5 |
| Food consumption score | Acceptable | 282 | 69.6 |
| | Borderline | 86 | 21.2 |
| | Poor | 37 | 9.1 |
| Nutritional status | Underweight | 68 | 16.8 |
| | Normal | 324 | 80.0 |
| | Overweight | 13 | 3.2 |

consumption pattern was significant, $x^2$ (1, N = 405) = 27.00, p < 0.001. Females (57.9%) were more likely to skip breakfast than were males (32.1%). Also, the chi-square test of independence for residence and breakfast consumption pattern was significant, $x^2$ (1, N = 405) = 73.92, p < 0.001. Being a rural resident (70.2%) was more likely to skip breakfast than were urban residents (27.3%) (Fig 1).

Among those who ate breakfast regularly, 67.4% ate it at home, with 55.1% of respondents' mothers preparing their breakfast. According to the report by breakfast skippers (188), the most common reason for skipped breakfast was "they did not have enough time (29.7%)" followed by "poor appetite in the morning" (17.6%)". The least common reason was (3.5%) "Don't like to eat early" (Table 4).

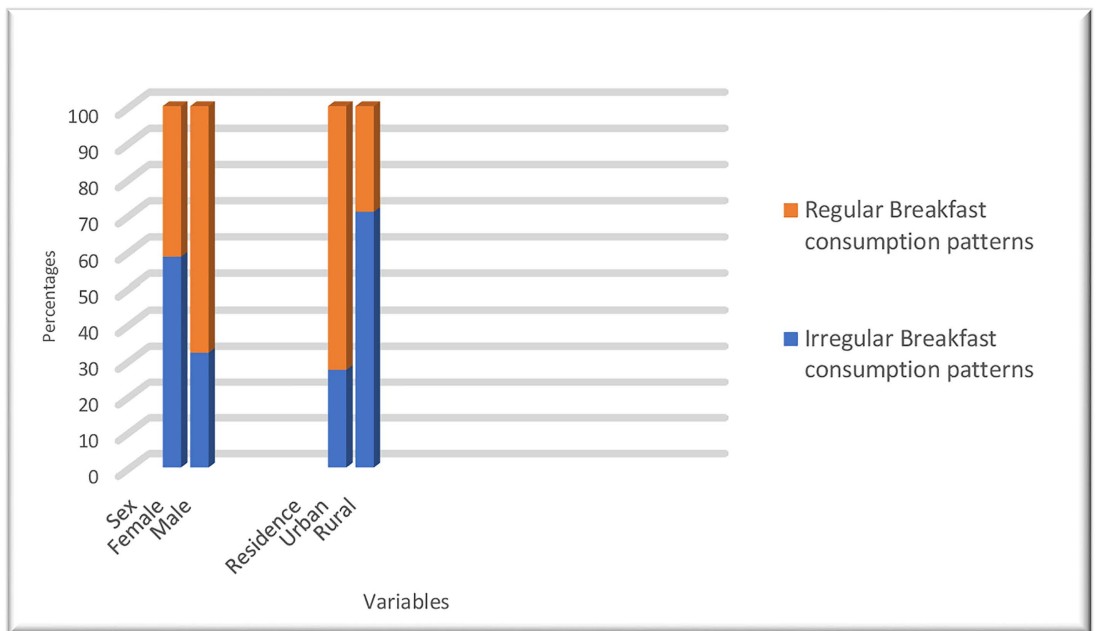

**Fig 1. Breakfast consumption patterns with sex and residence variables among adolescent high school students in Tullo district, Oromia Region, Eastern Ethiopia, 2023.**

### Factors associated with breakfast consumption patterns

In this study, variables such as sex, age, residence, family size, father's educational status, mother's educational status, father's occupation, average monthly income of the family, BMI for age, food consumption score, current chew khat, current cigarette smoke, time of wake-up, and having an eating disorder were associated with breakfast consumption patterns at a p-value of 0.25 in bivariable analysis to identify candidate variables for multivariable analysis. In multivariate analysis, however, only being female, having a family size of >5, being a rural resident, having no formal education from the respondent's mother, chewing khat, cigarette smoking, and eating disorders were significantly associated with breakfast consumption patterns among adolescents (Table 5).

Accordingly, female participants were 5.28 (AOR = 5.28; 95% CI: 2.69, 10.36) times more likely to skip breakfast than male participants. Participants who had > 5 family members were 4.76 (AOR = 4.76; 95% CI: 2.41–9.36) times more likely to skip breakfast than <5 family members. Rural residents were 3.34 (AOR = 3.34; 95% CI: 1.78, 6.25) times more likely to skip breakfast than urban residents. Participants' mothers who never attained formal education were 3.89 (AOR = 3.89; 95% CI: 2.09, 7.22) times more likely to skip breakfast compared to those who attained primary and above school. There was a 3.13 (AOR = 3.13; 95% CI: 1.59, 6.16) times greater chance of skipping breakfast among those who chewed khat in the past 30 days than those who did not chew khat, and smoking cigarettes was associated with 4.12 times (AOR = 3.06; 95% CI: 1.02, 9.17) higher likelihood to skip breakfast than non-smokers. Of the participants who reported having an eating disorder or having one or more symptoms, they were 6.54 (AOR = 6.54; 95% CI: 2.19, 19.43) times more likely to skip breakfast than those who reported not having an eating disorder (Table 5).

### Discussion

According to this study, 46.2% (95% CI: 41.5, 51.4) of adolescent high school students skip breakfast. Our study found a consistent prevalence of breakfast skippers among study participants compared to other studies conducted in

**Table 4. Breakfast characteristics of adolescent high school students in Tullo district, Oromia Region, Eastern Ethiopia, 2023.**

| Variables | Category | Frequency (n = 405) | Percentage (%) |
|---|---|---|---|
| Where do you usually eat your breakfast | From my home | 273 | 67.4 |
| | On the way to school | 107 | 26.4 |
| | Others* | 25 | 6.2 |
| Who prepares breakfast for you | Mother | 223 | 55.1 |
| | Myself | 104 | 25.7 |
| | Sister | 51 | 12.6 |
| | Others** | 27 | 6.7 |
| Families, teachers, or friends encourage you to have breakfast regularly. | Never | 148 | 36.5 |
| | Sometimes | 84 | 20.7 |
| | Often | 45 | 11.1 |
| | Always | 128 | 31.6 |
| Do you think it is important to eat breakfast? | Neutral | 44 | 10.9 |
| | Yes | 283 | 69.9 |
| | No | 78 | 19.3 |
| Reason for breakfast skipping | I don't feel hungry | 32 | 7.9 |
| | No time to have breakfast | 119 | 29.7 |
| | I don't find ready food to eat | 34 | 8.4 |
| | My family skip the breakfast and so I do | 32 | 7.9 |
| | Poor appetite in the morning | 88 | 21.9 |
| | I need to lose weight | 26 | 6.5 |
| | I don't like to eat early | 14 | 3.5 |
| | Don't have food to eat | 56 | 14.0 |

*From school, From relative's house, **Brother, Grandmother, Father, Aunts

Ethiopia: in the North Shewa, Oromia Region, 41.3% [13], and in the Shebedino District, Southern Ethiopia, 42.3% [4]. This finding is also consistent with a comprehensive study conducted in more than 35 countries and a study conducted in India, which revealed that 45% [20] and 47.7% [32] of adolescents did not consume breakfast, respectively. Similarly, a study conducted in selected regions in Lebanon reported that the prevalence of breakfast skipping was 42.8% [33]. Whereas this study found a relatively higher prevalence of breakfast skippers compared to studies conducted in Mekelle City, Ethiopia (2019) revealed that among public and private secondary schools, from 853 total participants, 244 (28.6%) did not eat breakfast daily; among this group, 36.6% and 20% reported that breakfast intake was not eaten daily, respectively [14] and in Nigeria, 23% [12] of them had not eaten breakfast daily; in Palestine, 38% [11] and, in Jordan, 18.5% [34].

Regarding reasons for skipping breakfast meals, the study revealed that more than one fourth (29.7%) of the students skipped breakfast because they "don't have time to eat breakfast, followed by (21.9%) "poor appetite in the morning. These common reasons for skipping breakfast have also been reported in previous studies conducted in Ethiopia and other countries [11,13,34]. The reason for this finding may be many students chew khat to study at night and are often rushing to class since they have little time for breakfast in the morning. It has been reported that skipping breakfast has been a means of saving time by most adolescents in the morning to get to school on time [35].

In this study, female participants were 5.28 times more likely to skip breakfast than male participants. This finding is in line with a study conducted in North Shewa, Ethiopia [13], and similarly other studies also revealed that female participants were more likely to skip breakfast than males [19,36]. This could be because of the females are probably more

**Table 5. Factors associated with breakfast consumption patterns in adolescent high school students in Tullo district, Oromia region, Eastern Ethiopia 2023.**

| Variable | | Breakfast consumption patterns | | COR (95%CI) | AOR (95%CI) | P-value |
|---|---|---|---|---|---|---|
| | | Irregular n (%) | Regular n (%) | | | |
| **Sex of student** | Female | 128(58.4) | 93(41.6) | 2.92(1.94–4.39) | 5.28(2.69–10.36) | <0.001** |
| | Male | 59(32.1) | 125(67.9) | 1 | 1 | |
| **Family size category** | >5 Persons | 162(63.9) | 93(36.1) | 8.71(5.29–14.35) | 4.76(2.41–9.36) | <0.001** |
| | ≤5 Persons | 25(16.7) | 125(83.3) | 1 | 1 | |
| **Residence of student** | Rural | 125(70.8) | 53(29.2) | 6.27(4.07–9.69) | 3.34(1.78–6.25) | <0.001** |
| | Urban | 62(27.3) | 165(72.7) | 1 | 1 | |
| **Mother's educational status** | No formal education | 145(62.4) | 89(37.6) | 5.00(3.23–7.75) | 3.89(2.09–7.22) | <0.001** |
| | Formal education | 42(24.6) | 129(75.4) | 1 | 1 | |
| **Current khat use** | Yes | 115(66.3) | 60(33.7) | 4.21(2.76–6.39) | 3.13(1.59–6.16) | <0.001** |
| | No | 72(31.3) | 158(68.7) | 1 | 1 | |
| **Current cigarette use** | Yes | 53(88.3) | 7(11.7) | 11.92(5.26–26.99) | 3.06(1.02–9.17) | 0.046** |
| | No | 134(38.8) | 211(61.2) | 1 | 1 | |
| **Eating\disorder** | Yes | 36(81.8) | 13(22.2) | 3.76(1.92–7.33) | 6.54(2.19.–19.49) | <0.001** |
| | No | 151(43.3) | 205(56.7) | 1 | 1 | |

**Significant at P-value < 0.05, COR = Crude Odd Ratio, AOR = Adjusted odd ratio, CI = Confidence interval.*

self-conscious about their weight and/or appearance, which makes them more prone to engage in weight-controlling activities like skipping breakfast [37]. Similarly, in some cultures, women are more likely to prioritize family needs over their own, leading to less attention to personal health and nutrition, including skipping breakfast and women are more likely to eat in response to emotional states and social contexts, which can lead to irregular meal patterns, including skipping breakfast [38,39].

Additionally, rural residents were 3.34 times more likely to skip breakfast as compared to those who lived in urban areas. This result is in line with a study conducted in Jenin governance, West Bank, where the participants who lived in rural areas had 1.26 times the risk of always skipping breakfast compared to those living in major cities [11]., This could be due to the reason that rural environments often have different cultural norms and attitudes towards food compared to urban areas, and in some urban settings, there may be a greater emphasis on regular mealtimes and the importance of nutrition, which could influence individuals' patterns and reduce the likelihood of skipping breakfast [37]. Additionally, rural students often face food insecurity and limited access to nutritious meals [40].

The current study also revealed that higher maternal education would improve regular breakfast consumption patterns. Participants whose mothers had no formal education were 3.89 times more likely to skip breakfast than those whose mothers had primary and above education. This result is in line with the study conducted in the Shebedino district, southern Ethiopia, and in Jidda, Saudi Arabia [19]. This is explained by mothers with no formal education, who may have limited knowledge about nutrition and healthy eating habits, and limited financial resources may result in food insecurity within the household, leading to breakfast skipping [41]. Additionally, higher maternal education levels lead to increased nutritional knowledge, enabling mothers to make informed dietary choices for their children and help them to understand the importance of regular breakfast consumption, leading to healthier eating habits in their children [42,43].

In this study, current smokers were 3.06 times more likely to skip breakfast compared to nonsmokers. This is consistent with the study conducted in Bangladesh [44]. A possible explanation for the correlation between smoking and skipping breakfast is that smoking can decrease appetite, leading to lower breakfast consumption [45]. However, our finding is

inconsistent with a study conducted to assess breakfast consumption and its socio-demographic and lifestyle correlates in school children in 41 countries participating in the HBSC study [37].

Moreover, in this study, current khat chewers are 3.13 times more likely to skip breakfast than those who do not chew khat currently. This result is supported by research done on factor analysis–eating patterns among khat chewers in Saudi Arabia [24]. The possible explanation for this could be khat is often consumed in social settings as a communal activity that can alter eating patterns, such as skipping meals like breakfast, and its normalization among peers, particularly in educational institutions, reinforces its consumption while overshadowing traditional meal times [46,47]. In addition to this, individuals who chew khat may be less likely to make healthy choices regarding their diet, including skipping breakfast, and chewing khat may disrupt daily routines, including regular meal times [24].

The other variable that had an association with breakfast consumption pattern was eating disorder. Furthermore, this study revealed that breakfast skipping was 6.54 times more common among those who reported having the risk of an eating disorder than those who did not have. This study was supported by research conducted in the USA and high-income countries, which describes that during adolescence, regular breakfast consumption was negatively associated with an eating disorder [41,48,49]. Which can be described as the likelihood of developing any form of eating disorder was diminished by eating breakfast [50]. The possible explanation for this is that adolescents at risk for eating disorders may skip breakfast as a means of calorie restriction or due to body dissatisfaction, and the disruption in family meal practices can also lead to increased breakfast skipping, particularly among adolescents who prioritize autonomy in food choices [51]

This study is not without limitations. Since some questions asked about events that occurred four weeks before the study period, recall bias might be introduced into study; for this, sufficient time to think and answer questions and used neutral interviewers. Social desirability bias was also another limitation, since some students might report what they think is socially acceptable or favorable rather than their true experiences. This may lead to overestimation or understatement of certain behaviors. Self-administered questionnaires were used and confidentiality was ensured to minimize this bias.

## Conclusion

The finding of this study showed that the prevalence of irregular breakfast consumption (breakfast skipping) among adolescent high school students was high. Being female, rural residency, no formal maternal education, smoking of cigarettes, khat chewing, and eating disorders were identified as factors associated with breakfast consumption patterns. Based on this findings, school-based health education initiatives highlighting the importance of regular breakfast intake, especially among girls and urban people is recommended. Interventions must focus on educating mothers and discouraging unhealthy behaviors like smoking and chewing khat. Additionally, schools should promote eating problem screening and support and coordinated implementation of these strategies can increase breakfast intake among adolescents, which may benefit their academic and health outcomes. Further research that uses mixed study designs is recommended to understand in depth the breakfast consumption patterns and the culture surrounding it, including perceptions towards breakfast consumption patterns among adolescents.

## Supporting information

**S1 File.**
(XLS)

## Acknowledgments

We would like to express our gratitude to the staff of Tullo District high school, study participants, data collectors and supervisors for their tremendous support and assistance during the data collection process.

## Author contributions

**Conceptualization:** Natnael Teferi, Dawit Firdisa, Tara Wilfong, Samrawit Berihun, Behailu Hawulte.

**Data curation:** Natnael Teferi.

**Formal analysis:** Natnael Teferi, Dawit Firdisa.

**Investigation:** Behailu Hawulte.

**Methodology:** Samrawit Berihun, Behailu Hawulte.

**Supervision:** Tara Wilfong.

**Writing – original draft:** Natnael Teferi, Dawit Firdisa.

**Writing – review & editing:** Natnael Teferi, Dawit Firdisa, Tara Wilfong, Samrawit Berihun, Behailu Hawulte.

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
