## [Decision Letter · Decision Letter 0]

24 Apr 2025

PONE-D-25-08283Breakfast Consumption Patterns and Associated Factors among Adolescent High-School Students in Tullo District, Eastern EthiopiaPLOS ONE

Dear Dr. Firdisa,

Thank you for submitting your manuscript to PLOS ONE. After careful consideration, we feel that it has merit but does not fully meet PLOS ONE’s publication criteria as it currently stands. Therefore, we invite you to submit a revised version of the manuscript that addresses the points raised during the review process.

We look forward to receiving your revised manuscript.

Kind regards,

Omnia Samir El Seifi, M.D., Ph.D.

Academic Editor

PLOS ONE

2. In the online submission form, you indicated that [The raw data supporting the conclusions of this article will be made available by the authors without undue reservation.].

Reviewers' comments:

Reviewer's Responses to Questions

**Comments to the Author**

1. Is the manuscript technically sound, and do the data support the conclusions?

Reviewer #1: Yes

Reviewer #2: Yes

Reviewer #3: Yes

2. Has the statistical analysis been performed appropriately and rigorously? 

Reviewer #1: Yes

Reviewer #2: Yes

Reviewer #3: Yes

3. Have the authors made all data underlying the findings in their manuscript fully available?

Reviewer #1: Yes

Reviewer #2: Yes

Reviewer #3: Yes

4. Is the manuscript presented in an intelligible fashion and written in standard English?

Reviewer #1: Yes

Reviewer #2: Yes

Reviewer #3: No

5. Review Comments to the Author

Reviewer #1: The study aimed to assess breakfast consumption patterns and their associated factors among

adolescent high school students in the Tullo district, Eastern Ethiopia.

Overall, the study achieved its objectives, the methods followed are sound and the results interpretation is good. There is few points to address in order to achieve perfection. in the discussion, part it would be interesting to compare findings of this study with different populations of different age groups. with population of different socio-economic background and discuss the difference or similarities. It would be also interesting to highlight in the discussion the implications of these findings. in other terms knowing these % and associations what do we conclude or how this will help us, policy makers and other stakeholders. Overall, what is the advantage of knowing these findings. It would be great if you can associate these data with the school grades to see if there is any impact of skipping breakfast on school performance as mentioned in the abstract. check the impact on health status, IQ. Schools should have records for the assessed population.

In the discussion and conclusion the authors used "we" rephrase in passive tense.

Table 1 Education level % there is a"the" to be deleted.

in the tables add ? when you are quantifying a question

Table 4 "myself" one word. add ?

Figure to change the format, letter font and size. they are very blurr

Reviewer #2: Dear Authors,

The manuscript entitled “Breakfast Consumption Patterns and Associated Factors among Adolescent HighSchool Students in Tullo District, Eastern Ethiopia, addresses an important public health issue in a relatively under-researched context—adolescent nutrition in rural Ethiopia. The focus on breakfast consumption among high school students provides valuable insights into adolescent dietary behavior, which has strong links to educational performance and health outcomes. The research objective is well-articulated and grounded in existing literature gaps. The authors made a strong case for the need to assess breakfast habits in the Tullo District due to a lack of local data.

1. There are numerous grammatical errors and awkward phrasings throughout the manuscript. For instance, phrases like “There is growing proof to recommend...” should be revised to “There is growing evidence suggesting...”.

2. A professional language edit is highly recommended to enhance readability and clarity.

3. There is an inconsistency in stating “being female, urban residency…” as risk factors in the conclusion when rural residency was actually associated with higher breakfast skipping. This needs correction to avoid misinterpretation.

4. While the study finds associations, the cross-sectional design precludes any causal inference. The authors should be more cautious in statements like "causes breakfast skipping" and instead say “is associated with.”

5. The discussion, though comparative, would benefit from a deeper theoretical exploration of why some factors (e.g., maternal education, khat use) are linked to breakfast skipping. Including more culturally grounded explanations could enrich the discussion.

6. The classification for BMI is inconsistent; underweight is described as < -2, and overweight as >1. This appears incorrect based on WHO BAZ classification. Please clarify or correct this to avoid misclassification bias.

7. Tables are overly long and could benefit from simplification and clearer formatting. Consider merging or streamlining some tables for better presentation.

8. Figures were not included in the review copy. Please ensure they are properly embedded and described within the manuscript.

9. Some of the citations are outdated or not peer-reviewed. The study would benefit from including more recent, high-impact studies, especially from Sub-Saharan Africa or other LMIC contexts.

10. The manuscript lacks consideration for other potential confounders such as dietary diversity, access to food, school schedule, or mental health beyond eating disorders.

11. Define all abbreviations at first use in the abstract and main text (e.g., AOR, FFQ, BAZ).

12. Clarify why certain cut-offs (like ≥6 times per week for regular breakfast consumption) were chosen.

13. The conclusion should be more concise and avoid repetition from earlier sections.

Reviewer #3: This is a cross-sectional study, conducted in Eastern Ethiopia with a sample of 405 adolescents aged 14-19 years, regarding breakfast eating patterns and its associated factors. Of participants, 46% skipped breakfast, called as “irregular breakfast consumption”, which was significantly associated with being female, having family size >5 members, living in the rural area, no formal maternal education, chewing khat, smoking and presenting eating disorders.

The study is well described, and detailed, the analysis seems adequate and well conducted. However, major revision is recommended to improve consistency. Tables should be revised to provide clearer information for the reader.

Recommendations:

English review is recommended.

- Abstract:

o Page 2, lines 22-24: “there is a dearth of evidence on the comprehensive understanding of the breakfast consumption patterns. This study aimed to assess breakfast consumption patterns”. Is this the main aim of the study? Throughout the manuscript there is much more emphasis on the associated factors analysis, besides, there seems to be a good body of evidence on the literature regarding breakfast consumption, how does this study advances in relation to the already existing evidence?

o Page 2, lines 28-29: the sentence “The data were entered into Epidata version 4.6 and exported to SPSS Statistics version 27.0.1 for analysis” is not essential for the abstract section and can be subtracted.

o Page 3, line 43: it is written “urban residency”, however in line 36 it was mentioned as rural residency. Furthermore, in the conclusion section of the abstract there is no need to cite all associated factors one by one, the authors can summarize findings and find the most important message of the study, highlighting why these results matter for the literature.

- Introduction:

o When citing existing evidence of other studies regarding associated factors to breakfast eating patterns, bring more context about the studies, where was it conducted, what is the sample size, the age of the individuals, the year the data were collected, and the magnitude of association being compared?

o Page 3, lines 52-53: “and it should guarantee a median of 20-25% of the energy consumed throughout the day” is this information relevant for the study aims and findings?

o Page 3, lines 56-57: “Moreover, during this period adolescents have the greatest total energy requirement compared to any age group (~2,420 kcal/day)” this information seemed a bit floaty from the rest of the paragraph, it would benefit from further exploring how this information is related to the importance of eating breakfast during this period of life.

o Page 3, line 58: in which countries were observed the lowest and highest breakfast skipping prevalence cited?

o Page 3, line 59: usually it is written as “low- and middle-income countries”.

o Page 4, lines 79-81: “Different scholars have intensively studied factors associated with breakfast eating habits and the nutritional status of adolescents in Ethiopia (4, 10, 11). However, there is a lack of clear data or insufficient studies on adolescent nutrition in particular breakfast consumption patterns” what is being considered as the difference between “breakfast eating habits” and “breakfast consumption patterns”? The phrasing is confusing and do not depict well enough how this study advances in regards of the existing literature on the topic.

o Page 4, lines 82-83: “Moreover, breakfast consumption patterns data was not included in the Ethiopia Demographic and Health Survey (21)” why is this information important?

- Methods

o Congratulate to the authors for the well described methods.

o Page 7, line 144: describe what the acronym “TEM” means.

o Page 9, line 176: describe what the acronym “VIF” means.

- Results

o Page 9, lines 185-186: “According to this study, 43.2% were currently khat chewers, 14.8% were cigarette smokers, and 5.9% consumed alcoholic drinks in the past 30 days” are these data from adolescents or the parents?

o Table 3: improvement in description of each variable and category is recommended. “Food frequency” of some specific food group, or meal frequency, for the category: 3 times a day, a week?

o Table 5: the table would benefit from providing an overall row about frequency and percentage of occurrence for each breakfast consumption pattern category.

- Discussion

o When citing existing evidence of other studies, bring more context about them, what is the study design, where was it conducted, what is the sample size, the age of the individuals, the year the data were collected, etc., so the reader can compare this study and the cited study?

o Page 13, line 276: there seems to be a typo.

6. PLOS authors have the option to publish the peer review history of their article (what does this mean? ). If published, this will include your full peer review and any attached files.

**Do you want your identity to be public for this peer review?** For information about this choice, including consent withdrawal, please see our Privacy Policy .

Reviewer #1: **Yes: ** Christelle Bou-Mitri

Reviewer #2: **Yes: ** Dr Manne Munikumar

Reviewer #3: No

---

## [Author Response · Author response to Decision Letter 1]

1 Jul 2025

Date: June, 2025

Dear editorial board members (s) of the PLOS ONE journal

The authors have been recalled to revise the manuscript entitled “Breakfast Consumption

Patterns and Associated Factors among Adolescent High-School Students in Tullo District,

Eastern Ethiopia” under the PLOS ONE journal, which was submitted for publication. So, we

received the editor's and reviewer's revision comments to improve the manuscript before its

publication.

Thank you for receiving comments, suggestions, directions, and questions from the editor and

reviewer. As we said, we are very happy to receive constructive and valuable comments that will

improve the manuscript. Accordingly, we have considered all the comments, questions, directions,

and suggestions and provided a point-by-point response letter.

Finally, we have submitted all the required documents in their revised form. We hope that we have

addressed all the suggestions, directions, and raised questions, and if you believe that the point(s)

are not addressed, please let us know.

Thank you very much to all editors (s), and reviewers

On behalf of the authors

Yours sincerely,

Correspondence author

Point-by-point response letter

Reviewer 1 comments

The study aimed to assess breakfast consumption patterns and their associated factors among

adolescent high school students in the Tullo district, Eastern Ethiopia.

Overall, the study achieved its objectives, the methods followed are sound and the results

interpretation is good. There is few points to address in order to achieve perfection.

1.In the discussion, part it would be interesting to compare findings of this study with different

populations of different age groups. with population of different socio-economic background and

discuss the difference or similarities.

Authors reply: Thank you very much for your important comment and suggestions on

improving the manuscript. Since skipping breakfast consumption is not significantly

associated with age variable we couldn’t discuss it with population age group.

2. It would be also interesting to highlight in the discussion the implications of these findings. in

other terms knowing these % and associations what do we conclude or how this will help us,

policy makers and other stakeholders. Overall, what is the advantage of knowing these findings.

Authors reply: Thank you very much for your important comment and suggestions on

improving the manuscript. Based on your comment, we made a correction. Look at revised

manuscript.

2.It would be great if you can associate these data with the school grades to see if there is any

impact of skipping breakfast on school performance as mentioned in the abstract. check the impact

on health status, IQ. Schools should have records for the assessed population.

Authors reply: Thank you very much for your important comment and suggestions on

improving the manuscript. Since skipping breakfast consumption is not significantly

associated with school grades, and school performance (IQ)we couldn’t discuss it.

3.In the discussion and conclusion, the authors used "we" rephrase in passive tense.

Authors reply: Thank you very much for your important comment and suggestions on

improving the manuscript. Based on your comment, we made a correction. Look at revised

manuscript.

4.Table 1 Education level % there is a"the" to be deleted. in the tables add? when you are

quantifying a question

Authors reply: Thank you very much for your important comment and suggestions on

improving the manuscript. Based on your comment, we made a correction. Look at revised

manuscript.

5.Table 4 "myself" one word. add ?

Authors reply: Thank you very much for your important comment and suggestions on

improving the manuscript. Based on your comment, we made a correction. Look at revised

manuscript.

6.Figure to change the format, letter font and size. they are very blurr

Authors reply: Thank you very much for your important comment and suggestions on

improving the manuscript. Based on your comment, we made a correction. Look at revised

manuscript.

Reviewer 2 comments

Dear Authors,

The manuscript entitled “Breakfast Consumption Patterns and Associated Factors among

Adolescent High School Students in Tullo District, Eastern Ethiopia, addresses an important public

health issue in a relatively under-researched context—adolescent nutrition in rural Ethiopia. The

focus on breakfast consumption among high school students provides valuable insights into

adolescent dietary behavior, which has strong links to educational performance and health

outcomes. The research objective is well-articulated and grounded in existing literature gaps. The

authors made a strong case for the need to assess breakfast habits in the Tullo District due to a lack

of local data.

Authors reply: Thank you very much for your POSITIVE and Energetic feedback the

manuscript.

1. There are numerous grammatical errors and awkward phrasings throughout the manuscript. For

instance, phrases like “There is growing proof to recommend...” should be revised to “There is

growing evidence suggesting...”.

Author's reply: Thank you very much for your important comment and suggestions on

improving the manuscript. Based on your comment, we corrected. Look at the revised

manuscript.

2. A professional language edit is highly recommended to enhance readability and clarity.

Author's reply: Thank you very much for your important comment and suggestions on

improving the manuscript. Based on your comment, we corrected. Look at the revised

manuscript.

3. There is an inconsistency in stating “being female, urban residency…” as risk factors in the

conclusion when rural residency was actually associated with higher breakfast skipping. This

needs correction to avoid misinterpretation.

Author's reply: Thank you very much for your important comment and suggestions on

improving the manuscript. Based on your comment, we corrected. Look at the revised

manuscript.

4. While the study finds associations, the cross-sectional design precludes any causal inference.

The authors should be more cautious in statements like "causes breakfast skipping" and instead

say “is associated with.”

Author's reply: Thank you very much for your important comment and suggestions on

improving the manuscript. Based on your comment, we corrected. Look at the revised

manuscript.

5. The discussion, though comparative, would benefit from a deeper theoretical exploration of

why some factors (e.g., maternal education, khat use) are linked to breakfast skipping. Including

more culturally grounded explanations could enrich the discussion.

Author's reply: Thank you very much for your important comment and suggestions on

improving the manuscript. Based on your comment, we corrected. Look at the revised

manuscript.

6. The classification for BMI is inconsistent; underweight is described as <-2, and overweight as

>1. This appears incorrect based on the WHO BAZ classification. Please clarify or correct this to

avoid misclassification bias.

Author's reply: Thank you very much for your important comment and suggestions on

improving the manuscript. Based on the WHO BAZ classification, it is a correct

classification using Z-score.

7. Tables are overly long and could benefit from simplification and clearer formatting. Consider

merging or streamlining some tables for better presentation.

Author's reply: Thank you very much for your important comment and suggestions on

improving the manuscript.

8. Figures were not included in the review copy. Please ensure they are properly embedded and

described within the manuscript.

Author's reply: Thank you very much for your important comment and suggestions on

improving the manuscript. Based on your comment, we corrected. Look at the revised

manuscript.

9. Some of the citations are outdated or not peer-reviewed. The study would benefit from

including more recent, high-impact studies, especially from Sub-Saharan Africa or other LMIC

contexts.

Author's reply: Thank you very much for your important comment and suggestions on

improving the manuscript. Based on your comment, we corrected. Look at the revised

manuscript.

10. The manuscript lacks consideration for other potential confounders such as dietary diversity,

access to food, school schedule, or mental health beyond eating disorders.

Author's reply: Thank you very much for your important comment and suggestions on

improving the manuscript.

11. Define all abbreviations at first use in the abstract and main text (e.g., AOR, FFQ, BAZ).

Author's reply: Thank you very much for your important comment and suggestions on

improving the manuscript. Based on your comment, we corrected. Look at the revised

manuscript.

12. Clarify why certain cut-offs (like ≥6 times per week for regular breakfast consumption) were

chosen.

Author's reply: Thank you very much for your important comment and suggestions on

improving the manuscript. Cut-offs like “≥6 times per week for regular breakfast

consumption” were used since, as a rule of thumb, at least one person should consume

breakfast six times per week to say no skipping.

13. The conclusion should be more concise and avoid repetition from earlier sections.

Author's reply: Thank you very much for your important comment and suggestions on

improving the manuscript. Based on your comment, we corrected. Look at the revised

manuscript.

Reviewer 3 comments

Reviewer #3: This is a cross-sectional study, conducted in Eastern Ethiopia with a sample of 405

adolescents aged 14-19 years, regarding breakfast eating patterns and its associated factors. Of

participants, 46% skipped breakfast, called as “irregular breakfast consumption”, which was

significantly associated with being female, having family size >5 members, living in the rural

area, no formal maternal education, chewing khat, smoking and presenting eating disorders.

The study is well described, and detailed, the analysis seems adequate and well conducted.

However, major revision is recommended to improve consistency.

Tables should be revised to provide clearer information for the reader.

Authors reply: Thank you very much for your important comment and suggestions on

improving the manuscript. Based on your comment, we made a correction. Look at revised

manuscript.

Recommendations:

English review is recommended.

Authors reply: Thank you very much for your important comment and suggestions on

improving the manuscript. Based on your comment, we made a correction. Look at revised

manuscript. - Abstract:

o Page 2, lines 22-24: “there is a dearth of evidence on the comprehensive understanding of the

breakfast consumption patterns. This study aimed to assess breakfast consumption patterns”. Is

this the main aim of the study? Throughout the manuscript there is much more emphasis on the

associated factors analysis, besides, there seems to be a good body of evidence on the literature

regarding breakfast consumption, how does this study advances in relation to the already existing

evidence?

Authors reply: Thank you very much for your important comment and suggestions on

improving the manuscript. Based on your comment, we made a correction. Look at revised

manuscript. This study is focused on breakfast consumption patterns and factors associated

with it.

o Page 2, lines 28-29: the sentence “The data were entered into Epidata version 4.6 and exported

to SPSS Statistics version 27.0.1 for analysis” is not essential for the abstract section and can be

subtracted.

Authors reply: Thank you very much for your important comment and suggestions on

improving the manuscript. Based on your comment, we made a correction. Look at revised

manuscript.

o Page 3, line 43: it is written “urban residency”, however in line 36 it was mentioned as rural

residency. Furthermore, in the conclusion section of the abstract there is no need to cite all

associated factors one by one, the authors can summarize findings and find the most important

message of the study, highlighting why these results matter for the literature.

Authors reply: Thank you very much for your important comment and suggestions on

improving the manuscript. Based on your comment, we made a correction. Look at revised

manuscript. - Introduction:

o When citing existing evidence of other studies regarding associated factors to breakfast eating

patterns, bring more context about the studies, where was it conducted, what is the sample size,

the age of the individuals, the year the data were collected, and the magnitude of association

being compared?

Authors reply: Thank you very much for your important comment and suggestions on

improving the manuscript.

o Page 3, lines 52-53: “and it should guarantee a median of 20-25% of the energy consumed

throughout the day” is this information relevant for the study aims and findings?

Authors reply: Thank you very much for your important comment and suggestions on

improving the manuscript. We hope this study is relevant for knowing how much the

breakfast consumption affects our performance if we miss it.

o Page 3, lines 56-57: “Moreover, during this period adolescents have the greatest total energy

requirement compared to any age group (~2,420 kcal/day)” this information seemed a bit floaty

from the rest of the paragraph, it would benefit from further exploring how this information is

related to the importance of eating breakfast during this period of life.

Authors reply: Thank you very much for your important comment and suggestions on

improving the manuscript. Based on your comment, we made a correction. Look at revised

manuscript.

o Page 3, line 58: in which countries were observed the lowest and highest breakfast skipping

prevalence cited?

Authors reply: Thank you very much for your important comment and suggestions on

improving the manuscript. Based on your comment, we made a correction. Look at revised

manuscript.

o Page 3, line 59: usually it is written as “low- and middle-income countries”.

Authors reply: Thank you very much for your important comment and suggestions on

improving the manuscript. Based on your comment, we made a correction. Look at revised

manuscript.

o Page 4, lines 79-81: “Different scholars have intensively studied factors associated with

breakfast eating habits and the nutritional status of adolescents in Ethiopia (4, 10, 11). However,

there is a lack of clear data or insufficient studies on adolescent nutrition in particular breakfast

consumption patterns” what is being considered as the difference between “breakfast eating

habits” and “breakfast consumption patterns”? The phrasing is confusing and do not depict well

enough how this study advances in regards of the existing literature on the topic.

Authors reply: Thank you very much for your important comment and suggestions on

improving the manuscript. Based on your comment, we made a correction. Look at revised

manuscript.

o Page 4, lines 82-83: “Moreover, breakfast consumption patterns data was not included in the

Ethiopia Demographic and Health Survey (21)” why is this information important?

Authors reply: Thank you very much for your important comment and suggestions on

improving the manuscript. Based on your comment, we made a correction. Look at revised

manuscript. - Methods

o Congratulate to the authors for the well described methods.

o Page 7, line 144: describe what the acronym “TEM” means.

Authors reply: Thank you very much for your important comment on improving the

manuscript. Based on your comment, we made a correction. Look at revised manuscript.

o Page 9, line 176: describe what the acronym “VIF” means.

Authors reply: Thank you very much for your important comment on improving the

manuscript. Based on your comment, we made a correction. Look at revised manuscript. - Results

o Page 9, lines 185-186: “According to this study, 43.2% were currently khat chewers, 14.8%

were cigarette smokers, and 5.9% consumed alcoholic drinks in the past 30 days” are these data

from adolescents or

---

## [Decision Letter · Decision Letter 1]

21 Jul 2025

Breakfast Consumption Patterns and Associated Factors among Adolescent High-School Students in Tullo District, Eastern Ethiopia

PONE-D-25-08283R1

Dear Dr. Firdisa,

We’re pleased to inform you that your manuscript has been judged scientifically suitable for publication and will be formally accepted for publication once it meets all outstanding technical requirements.

Kind regards,

Omnia Samir El Seifi, M.D., Ph.D.

Academic Editor

PLOS ONE

Additional Editor Comments (optional):

Reviewers' comments:

Reviewer's Responses to Questions

**Comments to the Author**

1. If the authors have adequately addressed your comments raised in a previous round of review and you feel that this manuscript is now acceptable for publication, you may indicate that here to bypass the “Comments to the Author” section, enter your conflict of interest statement in the “Confidential to Editor” section, and submit your "Accept" recommendation.

Reviewer #2: (No Response)

Reviewer #3: All comments have been addressed

2. Is the manuscript technically sound, and do the data support the conclusions?

Reviewer #2: Yes

Reviewer #3: Yes

3. Has the statistical analysis been performed appropriately and rigorously? 

Reviewer #2: Yes

Reviewer #3: Yes

4. Have the authors made all data underlying the findings in their manuscript fully available?

Reviewer #2: Yes

Reviewer #3: Yes

5. Is the manuscript presented in an intelligible fashion and written in standard English?

Reviewer #2: Yes

Reviewer #3: Yes

6. Review Comments to the Author

Reviewer #2: Dear Author,

Thank you for your responses regarding the manuscript entitled "Breakfast Consumption Patterns and Associated Factors among Adolescent High-School Students in Tullo District, Eastern Ethiopia." Based on its current form, the manuscript appears suitable for publication.

Reviewer #3: The study is well described, and detailed, the analysis seems adequate and well conducted. Authors responded sufficiently to previous comments and suggestions regarding improvements to the manuscript.

7. PLOS authors have the option to publish the peer review history of their article (what does this mean? ). If published, this will include your full peer review and any attached files.

**Do you want your identity to be public for this peer review?** For information about this choice, including consent withdrawal, please see our Privacy Policy .

Reviewer #2: No

Reviewer #3: **Yes: ** Caroline Zani Rodrigues

---

## [Editor Report · Acceptance letter]

PONE-D-25-08283R1

PLOS ONE

Dear Dr. Firdisa,

I'm pleased to inform you that your manuscript has been deemed suitable for publication in PLOS ONE. Congratulations! Your manuscript is now being handed over to our production team.

Kind regards,

on behalf of

Professor Omnia Samir El Seifi

Academic Editor

PLOS ONE